# Selenium Status and Oxidative Stress in SARS-CoV-2 Patients

**DOI:** 10.3390/medicina59030527

**Published:** 2023-03-08

**Authors:** Andrejs Šķesters, Anna Lece, Dmitrijs Kustovs, Maksims Zolovs

**Affiliations:** 1Scientific Laboratory of Biochemistry, Riga Stradins University, LV-1007 Riga, Latvia; 2Department of Pharmaceutical Chemistry, Riga Stradins University, LV-1007 Riga, Latvia; 3Statistics Unit, Riga Stradins University, LV-1007 Riga, Latvia; 4Institute of Life Sciences and Technology, Department of Biosystematics, Daugavpils University, LV-5401 Daugavpils, Latvia

**Keywords:** COVID-19, selenium, selenoprotein P, 4-HNE adducts, reactive oxygen species, oxidative stress

## Abstract

*Background and Objectives*: Insufficient intake of essential micronutrient selenium (Se) increases the susceptibility to diseases associated with oxidative stress. The study aim was to assess Se status and oxidative stress in COVID-19 patients depending on severity of the disease. *Materials and Methods*: Blood plasma of 80 post-COVID-19 disease patients and 40 acutely ill patients were investigated. Concentration of Se was detected by a fluorometric method with di-amino-naphthalene using acidic hydrolysis. Selenoprotein P (Sepp1), malondialdehyde (MDA), and 4-hydroxynonenal (4-HNE) and their metabolite adducts were evaluated by spectrophotometric methods using commercial assay kits. *Results*: Obtained results demonstrated that Se and Sepp1 concentration in acute patients were significantly (*p* < 0.05 for Se and *p* < 0.001 for Sepp1) decreased compared with post-COVID-19 disease patients. However, in post-COVID-19 disease patients, Se values were close to the low limit of the norm for the European population. 4-HNE adducts concentration as a marker of lipid peroxidation was significantly increased in the acute patients group compared to the recovery group (*p* < 0.001). *Conclusions*: COVID-19 pathology is characterized by the induction of oxidative stress and suppression of antioxidant defenses during the acute phase. Lower levels of Se and Sepp1 and higher levels of reactive oxygen species reflect this imbalance, highlighting the role of oxidative stress in the disease’s pathogenesis.

## 1. Introduction

Trace element selenium (Se) has attracted the attention of researchers over the past decades, but its true potential as a biologically active entity is far from clear understanding. Se performs its function via Se containing proteins which mostly contain selenocysteine at their active centre, and up to date, about 25 Se containing proteins have been identified in the cells of various organs and tissues in humans. Selenoproteins play a vital role in several physiological functions, such as thioredoxin reductases and deiodinases that regulate cellular redox homeostasis and thyroid metabolism. The glutathione peroxidase family of enzymes function as reactive oxygen species scavengers, and selenoprotein K contributes to immune cell activation and proliferation. Moreover, selenoprotein P (Sepp1) acts as a selenium transporter and represents the major selenoprotein in plasma, synthesized in the liver, secreted into the plasma, and transported to the target tissues [1]. Thus, Se plays a crucial role in the redox reactions and cell signalling, antioxidant processes, immune system [2], hormone metabolism, cardiovascular system [3], and maintenance of reproductive function [4]. Thus, it is not surprising that several studies have identified an association between Se depletion and diverse pathologies, such as cardiovascular diseases [5], heart failure [6], certain cancers [7], impaired immune function, infertility [8], cognitive decline [9], and increased susceptibility to RNA viral infections.

SARS-CoV-2, along with Ebola, human immunodeficiency virus type 1 (HIV-1), coxsackievirus, influenza virus, SARS-CoV, and MERS-CoV, is classified as a single-stranded linear RNA virus [10]. Studies have demonstrated that inadequate levels of Se promote the replication, mutations, and virulence of multiple RNA viruses [11,12,13]. The involvement of Se in both innate (macrophages and neutrophils) and adaptive (T and B cells) immune system components suggests that Se deficiency may impede the non-specific cell-mediated immune response and lead to impairments in T-cell lymphocyte activation, proliferation, and differentiation [2,14]. This, in turn, may enhance the susceptibility to infections, resulting in increased morbidity and mortality [15]. Epidemiological studies performed in various countries, with a primary focus on China, the origin of SARS-CoV-2, have demonstrated a close association between the body’s Se level and the incidence, severity, and prognosis of COVID-19 disease caused by the SARS-CoV-2 virus [16,17,18,19]. A recent comprehensive review by Fakhrolmobasheri et al. has revealed a strong correlation between lower Se and Sepp1 levels and unfavourable outcomes in COVID-19 patients. Additionally, Se concentrations in COVID-19 patients were significantly lower than those in healthy individuals [20]. Variability in the findings reported in the review may be attributed to the lack of consideration given to the initial Se levels in each country’s population and the presence or absence of regional Se deficiency in the soil. Lower Se and Sepp1 levels were reportedly detected in COVID-19 patients across various nations, such as Belgium’s intensive care unit short-stayers (7–11 days), Germany’s deceased cases compared to discharges, India’s male patients, Nigeria’s patients, Russia’s moderate to severe cases, South Korea’s severe cases, and Turkey’s pregnant women with COVID-19 [20].

The content of Se in human plasma can be estimated by determining the concentration of not only elemental Se but also Sepp1. Sepp1 is the major Se transporter in the bloodstream and is responsible for delivering more than 50% of the total Se content in human plasma [21,22]. Recent scientific studies have demonstrated the multifaceted roles of Sepp1 in the body. Sepp1 exhibits enzyme-like activity similar to that of glutathione peroxidase, enabling it to reduce phospholipid hydroperoxide in the presence of glutathione and thereby counteracting the effects of oxidative stress [17].

The current scientific literature suggests that viral infections are accompanied by escalated reactive oxygen species (ROS) generation, which can facilitate viral replication by exacerbating the infection [17]. ROS are highly reactive molecules that are produced by cells during normal metabolism and in response to various stressors, including viral infection. While low levels of ROS can have beneficial effects on the immune system, excessive ROS production can lead to oxidative stress, which can damage cells and contribute to inflammation and disease. The ROS with polyunsaturated fatty acids can lead to lipid peroxidation, which generates a wide variety of oxidation products. Lipid peroxidation is a chain reaction process that can occur in cell membranes, where polyunsaturated fatty acids are abundant. The degree of lipid peroxidation is often used as a reliable biomarker of oxidative stress mediated damage. The concentrations of secondary lipid peroxidation products, such as malondialdehyde (MDA) and 4-hydroxynonenal (4-HNE) and their metabolite adducts in different body fluids such as the whole blood, blood plasma, serum, and in other tissues, are generally used as an indicator of lipid peroxidation and markers of oxidative damage. Higher formation of MDA is associated with various diseases including hypertension, diabetes, and atherosclerosis [23]. 4-HNE acts both as a signalling molecule and as a cytotoxic product of lipid peroxidation causing long-lasting biological consequences. Once formed, 4-HNE can promote cell survival or death. If 4-HNE can be enzymatically metabolized at the physiological level, cells can survive. However, if 4-HNE is at a high or very high level, it induces apoptosis or necrosis-programmed cell death, eventually leading to molecular cell damage which may facilitate development of various pathological states [24].

The objective of this investigation was to evaluate Se status in COVID-19 patients during the acute phase and in recovered patients after 2 months, using Se and Sepp1 concentrations as biomarkers. We also aimed to determine the extent of oxidative stress by measuring MDA and 4-HNE adduct concentrations in plasma, with the intention of uncovering any potential relationship between Se status, disease severity, and oxidative stress.

## 2. Materials and Methods

The study was carried out on a cohort of 120 participants, comprising 40 acutely ill COVID-19 patients who were hospitalized in the COVID-19 unit or intensive care unit at Pauls Stradins Clinical University Hospital and 80 individuals who had recovered from COVID-19 and were discharged from the hospital 2 months earlier. The present research was conducted in compliance with the Declaration of Helsinki, Guidance on Good Clinical Practices, and applicable regulatory requirements. This clinical trial was conducted as a component of the Latvia Research Programme, which aims to investigate the clinical, biochemical, and immunogenetic paradigms of COVID-19 infection and their associations with socio-demographic, etiological, pathogenetic, diagnostic, therapeutic, and prognostic factors for inclusion in guidelines. The concentration of Se in blood plasma was evaluated using the fluorometric method with di-amino-naphthalene and acidic hydrolysis. The measurement of Sepp1 was carried out using the previously reported Human Sepp1 ELISA kit [25]. Chemicals were produced by Sigma-Aldrich (Spruce St., Saint Louis, MO, USA). MDA was detected using OxiSelect™ TBARS (MDA Quantitation) Assay kit, Cell Biolabs, Inc, San Diego, CA, USA, according to the manufacturer’s instruction. 4-HNE adducts were detected using OxiSelect™ HNE Adducts Competitive ELISA kit, Cell Biolabs, Inc, San Diego, CA, USA, according to the manufacturer’s instructions. The absorption spectrum was measured using multimodal microplate reader SPARK, TECAN, Austria. Blood samples were collected utilizing BD Vacutainers LH 1701.U Plus Blood Collection tubes, followed by centrifugation at a force of 1500× *g* and 4 °C for a duration of 10 min. The resulting blood plasma was fractionated into cryogenic tubes and then preserved by freezing at −80 °C prior to analysis.

Statistical analysis was performed using RStudio v.2021.09.1 (Boston, MA, USA) statistical software. Data distribution was checked visually (histograms and QQ-plots) and with Shapiro–Wilk test. MDA and 4-HNE adducts were log transformed due to positively skewed distributions. The Levene’s test was used to test if samples have equal variances. Statistical analysis of differences between groups was carried out using analysis of variance (ANOVA) and compared by the Tukey’s HSD test. *p*-value < 0.05 was set as statistically significant level. Data are presented as mean ± standard deviation and graphically.

## 3. Results

The study included 120 patients who met the eligibility criteria for analysis. Blood plasma samples were collected from all patients, and their Se, Sepp1, MDA, and 4-HNE adducts status were assessed. The patients were divided into three groups based on their COVID-19 infection status: infected in the spring–early summer period (March-June; Group 1) and in the autumn period (October; Group 2) both two months post-discharge from the hospital, and acutely ill patients (March-June; Group 3), divided into two subgroups: patients treated in the COVID-19 unit and patients treated in the intensive care unit (20 + 20). A previous study in healthy adults (*n* = 195, mean age 49, range 21–59) was utilized to obtain the average concentrations of Se, Sepp1, MDA, and 4-HNE adducts in the population [26]. The results that were obtained are presented in Figure 1 and Figure 2 below.

The Se content in the blood plasma of acutely ill patients at the hospital was significantly decreased (*p* < 0.05) compared to post-COVID-19 disease patients, as shown in Figure 1a. Furthermore, an even greater decrease in Se concentration was observed when samples from patients treated in intensive care units were analysed separately, with their plasma containing only 59.3 μg/L of Se (15% lower compared to the whole group) and having an extreme range of limits (min/max) from 75.4 μg/L to 43.2 μg/L. In contrast, the Se content in the blood plasma of post-COVID-19 disease patients was close to the lower limit of the normal range for the European population, with no significant differences observed between the spring–summer (1st group) and summer–autumn (2nd group): 84.6 ± 20.7 μg/L and 88.2 ± 27.2 μg/L, respectively (Figure 1a). The Sepp1 concentration in plasma was measured to be 4.5 ± 2.4 mg/mL in acute patients, 5.5 ± 2.2 mg/mL in the 1st group of post-COVID-19 patients, and 6.8 ± 2.3 mg/mL in the 2nd group. A marked reduction in the concentration was noted in the acute patient group (*p* < 0.001) compared to the Se concentration in this group. Interestingly, the summer–autumn recovery group showed a significant increase in Sepp1 concentration compared to each other group, as shown in Figure 1b.

The levels of MDA were increased in all groups of patients studied, with concentrations exceeding those observed in healthy individuals: 26.6 ± 10.8 μmol/L, 31.0 ± 18.6 μmol/L, and 26.6 ± 10.8 μmol/L in acute patients, the spring–summer group, and the summer–autumn group, respectively. Analysis of the logarithmic-transformed data showed no statistically significant differences among the groups (refer to Figure 2a). A different pattern emerged when investigating the plasma concentration of 4-HNE adducts. The highest level was detected in acute patients, while the concentration of 4-HNE was significantly (*p* < 0.001) lower in the spring–summer group, but without a significant difference in the summer–autumn group: 5.1 ± 2.4 μg/L, 3.4 ± 1.9 μg/L, and 3.9 ± 1.8 μg/L, respectively. A similar trend was evident when examining the logarithmic-transformed data: 1.5 ± 0.5 vs. 1.1 ± 0.5 vs. 1.3 ± 0.4 (see Figure 2b). No correlation between the examined parameters was detected.

The study findings indicate significant differences between acute patients and post-COVID-19 recovery groups. The concentrations of Se and Sepp1 were markedly lower in the acute patients, in comparison to healthy individuals. Conversely, the oxidative stress biomarkers, MDA and 4-HNE adducts, were elevated beyond the acceptable range.

## 4. Discussion

Se is an essential micronutrient that has antioxidant properties and plays a role in immune function, while Sepp1 participates in regulating Se metabolism and transporting Se throughout the body. Therefore, it has been hypothesized that deficiency of Se and Sepp1 might enhance the susceptibility and severity of COVID-19.

Several studies confirmed a decrease in Se and associated components in patients with COVID-19 disease [27,28,29,30]. The results of our study indicate that hospitalized COVID-19 patients had reduced levels of Se in their blood plasma, which were lower than the European healthy adult reference ranges of 70–130 μg/L (median 87 μg/L) for Se concentration, with concentrations below 70 μg/L considered deficient [6]. In addition, the Se concentrations were also lower than the range of 77.4–81.5 µg/L (median 79.5 μg/L) observed in healthy individuals from Central Europe [31] and the reference data from a European cross-sectional analysis of serum Se, with a concentration of 84.4 ± 23.4 µg/L [16].

A similar outcome is seen with Sepp1 concentration in hospitalized patients, which was near the lower limit of adequate level of healthy Europeans (2.56–6.63 mg/L). These results are in line with other investigators who reported significantly lower Se levels in COVID-19 patients compared to healthy controls [16,18,19,20] not depending on the relatively high [32] or low [16,18,19] baseline Se level. A notable association has been shown between low Se level and COVID-19 disease incidence [18,19], severity [33], and outcomes [34], especially in hospitalised patients [16] and patients in intensive care unit, which is in conformity with our results and specify Se deficiency as risk factor for COVID-19 disease.

Comparable findings have been elaborated by Majeed et al., where the authors established that viral load could damage critical organs in the body, especially after interacting with enzymes such as angiotensin [35]. Important considerations can now be directed to supportive nutrient therapies such as Se to mitigate the susceptibility and long-term impacts of COVID-19 disease. Se is effective in improving the immune system and controlling inflammation that could result from the spread of the virus [30]. Therefore, the inability to overcome oxidative changes can largely result in negative outcomes such as fatalities.

In another study by Tsermpini and colleagues, the researchers demonstrated that SAR-COV-2 is connected to increased oxidation stress, which is an imbalance between the production of antioxidant systems and free radicals [36]. As a result, such impacts can lead to severe conditions of COVID-19 disease because of their participation in the pathogenesis. With the increase in oxidative stress, various mechanisms such as DNA damage, protein inactivation, and lipid peroxidation can occur, which can potentially cause cell dysfunctions or result in an inflammatory response. Interestingly, Muhammad et al. found significant depletion of antioxidant levels in COVID-19 patients, resulting in increased formation of free radicals [27,37]. For instance, the researchers noted that essential elements such as Se, zinc, copper, and manganese were lower among patients with COVID-19 compared to controls. Similarly, plasma-concentrated nutrients such as vitamin A, C, and E were also lower in COVID-19 patients.

In another study, to determine the relationship between COVID-19 and oxidative stress (vascular stress and lipid peroxidation), the authors demonstrated that there was a reduction in antioxidant capacity, and thus, the patients were unable to counteract oxidative modifications [37,38]. Moreover, our results are consistent with previous studies indicating that oxidative stress markers are significantly elevated in COVID-19 patients. The significantly higher concentrations of 4-hydroxy-2-nonenal (4-HNE) protein adducts in acutely ill (hospitalized) patients suggest impaired antioxidant defense mechanisms in these individuals. Similarly, Mehri et al. reported increased malondialdehyde (MDA) levels in intensive care unit patients, accompanied by heightened enzymatic antioxidant activity [39]. The reduction in 4-HNE adduct concentrations two months after discharge suggests an improvement in dysregulated antioxidant system response to oxidative stress. Nevertheless, our assessment of lipid peroxidation by MDA measurements indicates consistent increase in MDA levels across all patient groups, including those who have recovered, which is in line with previous studies. Ikonomidis and co-workers reported elevated MDA level compared to controls at 4 months post-infection that remained increased even at 12 months after COVID-19 infection, when disease symptoms almost disappeared [40]. Another research group detected considerably higher MDA levels in COVID-19 patients 4 months after infection not associated with disease severity (mild, moderate, and severe) [41]. Thus, it can be considered that COVID-19 patients have elevated oxidative stress extending over a long post-infection period. Additionally, oxidative stress may play a significant role in the mechanism of disease and complications, especially endothelial and vascular development. Therefore, patients with COVID-19 may benefit from strategies to reduce or prevent oxidative stress.

Our results show that Se status among observed COVID-19 patients in Latvia, as assessed by Se and Sepp1 concentration in human plasma, is low among acute patients and patients 2 months following their discharge from the hospital. Therefore, current results are consistent with previous studies showing that Se deficiency is a global problem [15]. The deficiency of Se can result from insufficient dietary intake or increased requirements. The observed patients are affected by both factors, as the soil in Latvia, along with other regions in Eastern Europe, lacks sufficient Se content. This, in turn, affects the Se levels in plants, grain products, and other representatives of the food chain [42]. On the other hand, RNA viral infections can increase the production of ROS, which can lead to oxidative stress and damage to cells and tissues. In response to this, the host antioxidant defence system is activated, and the immune system is also stimulated to fight the infection. Selenoproteins play an important role in both the antioxidant defence system and the immune system [12]. Moreover, COVID-19 infection can lead to inflammatory and hypoxic conditions that may increase the need for Se and exacerbate Se deficiency. Considering all that is mentioned above, there is extraordinary question regarding the use of Se supplementation for COVID-19 patients. In terms of the type of Se compound that is more suitable for supplementation, organic Se compounds such as selenomethionine and selenocysteine are generally considered to be more bioavailable and better absorbed than inorganic forms of Se such as sodium selenite [43]. However, the optimal form and dose of Se for COVID-19 patients has not been established and requires further research. Oral supplementation is the most common form of Se supplementation and may be sufficient for individuals with mild Se deficiency. However, individuals with more severe Se deficiency or those who are unable to tolerate oral supplementation may require intravenous administration. The optimal duration and dose of Se supplementation for COVID-19 patients also requires further investigation. Excessive Se intake can be toxic and lead to adverse health effects; therefore, the dose of Se should be carefully monitored and tailored to individual patient needs [44].

While the role of Se supplementation in the treatment of COVID-19 is an area of active research, there is currently limited evidence to support its routine use in clinical practice. Any decisions regarding the use of Se supplementation should be made in consultation with a healthcare provider and based on individual patient needs.

### Strengths and Weaknesses of the Study

This research is critical because it helps to understand the interventions that can be incorporated to reduce adverse outcomes, such as the potential of developing other adverse conditions during high viral load, such as in the case of COVID-19. For instance, Žarković suggested how to respond to patients with SARS-CoV-2 by administering antioxidants such as lipophilic agents that can help to counter the spread of the virus [38]. Furthermore, Chavarría et al. established that antioxidant therapy can help to reduce the severity of conditions such as COVID-19 because it improves the survival rates of critical outcomes such as sequential organ failure assessment (SOFA) [27]. Similarly, the study by Molteni established that antioxidant molecules are effective in reducing the prognosis and symptoms of viral diseases such as COVID-19 [28]. There is no doubt that it is important to ensure redox balance by involving vital molecules to control the adverse effects of the virus. The presence of oxygen and nitrogen radicals should be regulated to prevent high concentration or low concentrations. It is worth noting that imbalances can result in a significant increase in viral load. However, the corrective measures may not be effective when patients have comorbidities such as diabetes, hypertension, and other diseases, which can largely cause oxidative stress [37]. Furthermore, while clinicians decide to use antioxidants in the treatment, it is important to ensure they are regulated and taken in their right dosages to reduce negative outcomes such as increased mortality and toxicity. Therefore, although there has been an elevated decline in critical nutrients such as Se in COVID-19 patients, it is important to understand the application of the research in clinical practice, such as the modifications that can be implemented following an increase of virus and the reduction of nutrients. Conversely, it is worth noting that the present study was conducted using a smaller sample size, and hence the results cannot be applied in other contexts without modification. Despite this limitation, the study can form a rich body of knowledge for researchers developing interventions for COVID-19 disease [30]. For example, paying attention to the trace element (Se) can help in reducing adverse outcomes such as an increase in the spread of infections during crises such as the COVID-19 pandemic. Therefore, clinicians can use the knowledge acquired in this review paper to develop mechanisms for controlling Se deficiency, which is associated with the development of pathogens [35]. This will help in reducing the mortality rate among patients with COVID-19 and other serious viral diseases. For instance, Alshammari et al. suggested the potential of developing Se-based dosages and compositions, such as inhalers and sprays, to control the spread of the COVID-19 pandemic [45].

## 5. Conclusions

Based on our results, it can be concluded that severe COVID-19 patients experience lower Se and Sepp1 concentrations and increased oxidative stress, which indicates heightened reactive oxygen species formation in the body. At two months post-infection, Se status appears to improve, with Se and Sepp1 concentrations reaching average population values for these biomarkers. Despite this, oxidative stress remains high, suggesting ongoing ROS formation and dysfunction of the antioxidant system, which are closely related to selenoproteins. Understandably, decrease in mortality rate among patients with COVID-19 can be controlled by regulating the oxidative stress biomarkers. While incorporating Se in the management of COVID-19 disease, it is important to consider the risks of toxicity, especially when injected or used as a therapy.

## Figures and Tables

**Figure 1 medicina-59-00527-f001:**
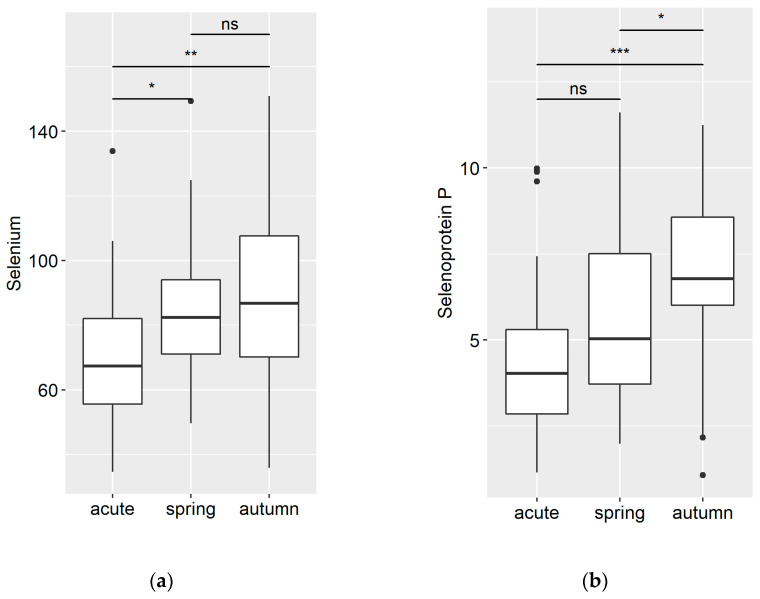
(**a**) Se concentrations, μg/L, and (**b**) Sepp1 concentrations, mg/mL, in blood plasma of acutely ill COVID-19 patients (acute) and in post-COVID-19 disease patients 2 months following their discharge from the hospital: infected with COVID-19 in the spring–early summer period (spring) or in summer–autumn period (autumn). Black dots show not significant outliers. * *p* < 0.05; ** *p* < 0.01; *** *p* < 0.001; ns—non-significant.

**Figure 2 medicina-59-00527-f002:**
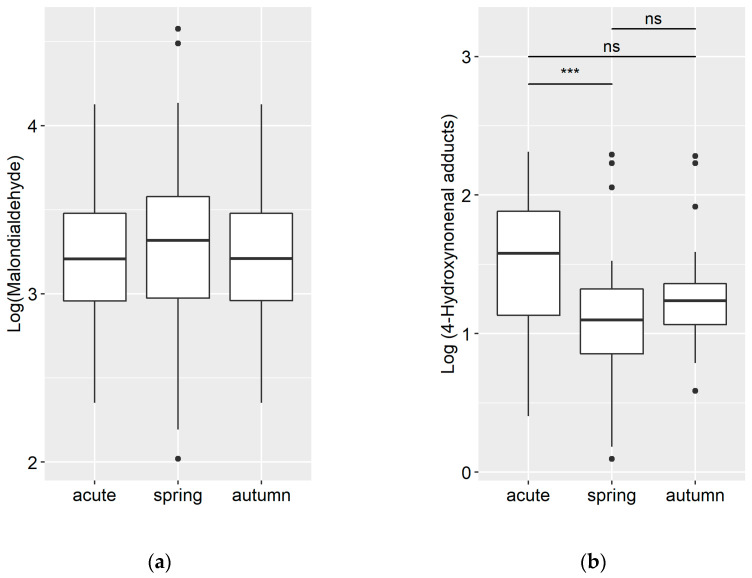
(**a**) MDA concentrations, μmol/L, and (**b**) 4-HNE adducts concentrations, μg/L, in blood plasma of acutely ill COVID-19 patients (acute) and in post-COVID-19 disease patients 2 months following their discharge from the hospital: infected with COVID-19 in the spring–early summer period (spring) or in summer–autumn period (autumn). Black dots show not significant outliers. *** *p* < 0.001; ns—non-significant.

## Data Availability

Not applicable.

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
