# Peer review of "Selenium Status and Oxidative Stress in SARS-CoV-2 Patients"

_medicina, 2023, doi:10.3390/medicina59030527_

Round 1

Reviewer 1 Report

This is an excellent paper, with clear presentation of obtained results and discussed in the light of general knowledge on COVID and Se role in viral diseases. My first impression was that the introduction was too long, however, after reading several times the entire manuscript, I believe it can stay in the present form. 

My request for changing- Starting the sentences with abbreviations should be highly discouraged, and please make the necessary changes in order to avoid this. This includes starting the sentence with Se.

My questions-

1. Why were the patients separated into two groups a.spring and b.autumn? Is this simply because of time of collecting the blood samples? Or is there any correlation of Se parameters and season?

2. Were the patients asked for Se supplementation? E.g. In my county, during the highest rates of COVID infection, the use of vitamin, mineral and herbal supplement simply skyrocketed. 

An advice- not mandatory- it would be useful to list advantages and disadvantages of the study, even if it was a single sentence in section Discussion.

Author Response

Dear Reviewer,

We appreciate the time you committed to providing us with detailed and constructive feedback. Your comments were helpful and we made certain corrections. 

Please find enclosed letter with corrections.

Sincerely,

Dmitrijs Kustovs

Reviewer 2 Report

This manuscript discussed the selenium status and oxidative stress in COVID-19 patients depending on the severity of the disease. The authors clearly demonstrated that COVID-19 involves the induction of antioxidant systems in acute disease, with patients having significantly lower concentrations of Se and SelP and a higher level of oxidative stress, which confirms the more intense formation of reactive oxygen species in the body.

I have only some suggestions:

1.     The manuscript should be thoroughly revised for spelling, grammar, and language structure.

2.     A more short introduction that focuses on context is required.

3.     The timing of the selection of post-COVID-19 patients should be justified.

4.     In statistical analysis section, addition of some details about statistics used should be done.

Author Response

(The authors gave the same response as above.)
